# SafeMERGE: Preserving Safety Alignment in Fine-Tuned Large Language Models via Selective Layer-Wise Model Merging

**Aladin Djuhera[1], Swanand Ravindra Kadhe[2], Farhan Ahmed[2], Syed Zawad[2], Holger Boche[1]**
[1] Technical University Munich, Chair of Theoretical Information Technology [2] IBM Research

## Abstract

Fine-tuning large language models (LLMs) on downstream tasks can inadvertently erode their safety alignment, even for benign fine-tuning datasets. We address this challenge by proposing SafeMERGE[1], a post-fine-tuning framework that preserves safety while maintaining task utility. It achieves this by selectively merging fine-tuned and safety-aligned model layers *only* when those deviate from safe behavior, measured by a cosine similarity criterion. We evaluate SafeMERGE against other fine-tuning- and post–fine-tuning-stage approaches for Llama-2-7B-Chat and Qwen-2-7B-Instruct models on GSM8K and PubMedQA tasks while exploring different merging strategies. We find that SafeMERGE consistently reduces harmful outputs compared to other baselines without significantly sacrificing performance, sometimes even enhancing it. The results suggest that our selective, subspace-guided, and per-layer merging method provides an effective safeguard against the inadvertent loss of safety in fine-tuned LLMs while outperforming simpler post–fine-tuning-stage defenses.

## 1 Introduction

Large language models (LLMs) have demonstrated remarkable capabilities in text generation and understanding while becoming increasingly accessible to AI practitioners. Safety tuning is critical to ensure that advanced LLMs align with human values and security policies, making them safe for deployment (Ouyang et al., 2022; Bai et al., 2022; Chiang et al., 2023; Zhang et al., 2024). However, the safety alignment of current LLMs has been shown to be vulnerable (Wei et al., 2023; Huang et al., 2024e; Yang et al., 2023; Zeng et al., 2024; Zhan et al., 2024; Qi et al., 2023; 2024a). In fact, fine-tuning LLMs on benign data (without any harmful content) can inadvertently degrade their previously established safety alignment (Qi et al., 2023). Since fine-tuning is pivotal for adapting generalist models to specialized tasks, ensuring that LLMs *remain safe after fine-tuning* is a critical practical challenge.

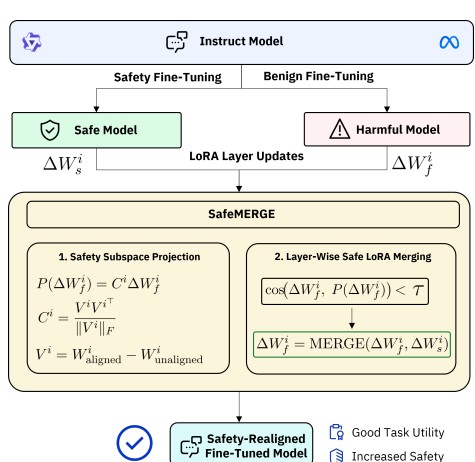

Figure 1: SafeMERGE merges harmful and safe LoRAs if the layers deviate from safe behavior, measured by a projection-based cosine similarity.

Recent defenses for preserving safety after fine-tuning can be divided into three categories based on the stage at which the solution is implemented: alignment-stage defenses (Huang et al., 2024d; Rosati et al., 2024), fine-tuning-stage defenses (Bianchi et al., 2024; Qi et al., 2024a; Huang et al., 2024c), and post–fine-tuning-stage defenses (Bhardwaj et al., 2024; Hsu et al., 2025). See (Yao et al., 2024) for a survey, with additional details in Appendix A. However, most of these works

---

[1]Code available at: https://github.com/aladinD/SafeMERGE

propose *custom* alignment or fine-tuning algorithms as defenses, which are often complex and require specialized knowledge for their implementation. This makes seamless integration with popular open-source libraries, such as Llama-Factory (Zheng et al., 2024) and Unsloth (Daniel Han & team, 2023), difficult, thereby hindering their adoption in practice. Moreover, existing defenses that do not require tailor-made alignment or fine-tuning algorithms often sacrifice task-specific utility in favor of preserving safety. Motivated by these practical challenges, we pose the following question:

*How can we help practitioners achieve task-specific utility while mitigating safety degradation when fine-tuning LLMs on downstream datasets, so that they can utilize popular open fine-tuning libraries without requiring custom fine-tuning and/or alignment techniques?*

In this paper, we address the above question by proposing **SafeMERGE**, a model-agnostic framework that leverages model merging (Ilharco et al., 2023; Matena & Raffel, 2022; Choshen et al., 2022; Yadav et al., 2023b; Yang et al., 2024b) to fuse the parameters of multiple fine-tuned models into a single model with combined capabilities. Figure 1 provides an overview of SafeMERGE. Beyond fine-tuning an LLM on task-specific data, SafeMERGE also fine-tunes the LLM on a small amount of safety-aligned data (e.g., harmful prompt-safe response pairs) to obtain a *safe model*. Inspired by (Hsu et al., 2025), SafeMERGE incorporates a *safety-aligned subspace* which is computed as the weight difference between the base (unaligned) and aligned versions of the model (e.g., Llama-2-7B and Llama-2-7B-Chat). This subspace of inner products of all possible weights represents the safety-related concept in aligned models, and can be used to identify *task vectors* that are harmful (Hsu et al., 2025). For each layer in the fine-tuned LLM, if the deviation from this safety-aligned subspace is large, SafeMERGE merges the corresponding layer from the fine-tuned model with that of the safe model, ensuring safety alignment while preserving task performance.

We evaluate SafeMERGE on two widely used LLMs, Llama-2-7B-Chat (Touvron et al., 2023) and Qwen-2-7B-Instruct (Yang et al., 2024a), across two benchmark tasks: GSM8K (Cobbe et al., 2021) and PubMedQA (Jin et al., 2019). We demonstrate that SafeMERGE significantly mitigates safety degradation after fine-tuning while preserving strong downstream task performance. Specifically, SafeMERGE achieves a better trade-off between utility and safety compared to existing baselines that similarly do not require custom fine-tuning or alignment. Additionally, we conduct several ablation studies to investigate key components, including different merging strategies, weighting schemes, and similarity thresholds.

## 2  SafeMERGE: Selective Layer-Wise Safe LoRA Model Merging

Given an aligned model and a task-specific dataset, our goal is to fine-tune the model to maximize task utility while minimizing safety degradation. We focus on efficient LoRA fine-tuning (Hu et al., 2021) which is widely used in practice. SafeMERGE achieves this goal by constructing (i) a *fine-tuned model*, obtained by fine-tuning on task-specific data, and (ii) a *safe model*, obtained by fine-tuning on safety-aligned data (e.g., harmful prompt-safe response pairs; see Appendix B.2).

Similarly to Hsu et al. (2025), SafeMERGE then uses the *safety-aligned subspace* to determine which layers of the fine-tuned model have been updated in a way that boosts utility but compromises safety. Let $\Delta W_f^i$ and $\Delta W_s^i$ denote the LoRA updates of the $i$-th layer for the fine-tuned and safe models, respectively, and let $W_{\text{aligned}}^i$ and $W_{\text{unaligned}}^i$ represent the weights of the $i$-th layer for the safety-aligned (e.g., instruct) and unaligned (e.g., base) models. The safety-aligned subspace for the $i$-th layer is then computed as $C^i = \frac{V^i V^{i^\top}}{\|V^i\|_F}$, where $V^i = W_{\text{aligned}}^i - W_{\text{unaligned}}^i$.

As shown in Hsu et al. (2025), a smaller cosine similarity between the fine-tuned and projected LoRA weights, $\Delta W_f^i$ and $C^i \Delta W_f^i$, indicates a greater deviation from the safety-aligned subspace. This observation allows us to identify layers with safety degradation as follows. Let $\rho^i$ denote the cosine similarity between $\Delta W_f^i$ and $C^i \Delta W_f^i$. Given a safety threshold $\tau \in [0, 1]$, $\rho^i \geq \tau$ indicates that $\Delta W_f^i$ is sufficiently safe, whereas $\rho^i < \tau$ signifies that $\Delta W_f^i$ has undergone safety degradation.

For each layer $\Delta W_f^i$ that has undergone safety degradation, SafeMERGE merges it with the corresponding layer $\Delta W_s^i$ from the safe model, yielding $\Delta W_{\text{merge}}^i = \text{MERGE}(\Delta W_f^i, \Delta W_s^i)$, where $\text{MERGE}(\cdot)$ defines the merging strategy. One example is linear merging with $\alpha \in [0, 1]$:

$$\Delta W_{\text{merge,linear}}^i = \alpha \Delta W_f^i + (1 - \alpha) \Delta W_s^i. \tag{1}$$

Note that the threshold $\tau$ controls the selectiveness of the merging approach, where a larger $\tau$ merges all layers, recovering a full (e.g., linear) combination of $\Delta W_f$ and $\Delta W_s$, while a smaller $\tau$ retains more fine-tuned updates. We present the impact of tuning $\tau$ in Appendices D.3.2 and D.3.3.

The key distinction of SafeMERGE is that, after identifying unsafe layers via the safety-aligned subspace, it *merges* them with safe layers. In contrast, SafeLoRA *projects* unsafe layers onto the safety-aligned subspace. While in both cases the safety-aligned subspace effectively identifies unsafe layers, we posit that merging them with safe layers achieves a better balance between utility and safety rather than simple projection. Our empirical evaluation supports this hypothesis, demonstrating the superiority of SafeMERGE over SafeLoRA.

## 3 EXPERIMENTAL SETUP

**Models and Datasets.** We LoRA fine-tune two widely used aligned models, Llama-2-7B-Chat and Qwen-2-7B-Instruct. Our main *utility datasets* are GSM8K (Cobbe et al., 2021), a math reasoning corpus with grade-school problems commonly used to benchmark multi-step reasoning, and PubMedQA (Jin et al., 2019), a biomedical corpus that tests domain-specific knowledge and medical context safety. Compared to GSM8K, it contains substantially more samples, offering a broader domain shift. Additionally, SafeMERGE requires a *safe model* for merging. We obtain this by fine-tuning the aligned model on subsets (100, 500, 1000, 2500 samples) of the safety data from Bianchi et al. (2024), selecting the safest variant. We provide more fine-tuning details in Appendix B.

**Evaluation Setup.** To assess task performance on the *utility datasets*, we report exact-match accuracy for GSM8K and classification accuracy for PubMedQA. For *safety evaluations*, we follow Yao et al. (2024); Qi et al. (2024a); Hsu et al. (2025) and generate responses on DirectHarm (Lyu et al., 2024) and HexPhi (Qi et al., 2024b), which contain harmful prompts that conflict with aligned LLM policies. We use Llama-Guard-3-8B (Llama Team, 2024) to judge their harmfulness and measure safety as the harmful output rate (lower is better). We exclude AlpacaEval (Li et al., 2023) from our evaluation, as our primary concerns are safety and utility in multi-step reasoning or domain QA tasks rather than general instruction following as in (Hsu et al., 2025). We provide details in Appendix C.

**Baselines.** We compare SafeMERGE against baselines that align with our goal of *post-hoc safety corrections*, i.e., methods that do not require custom fine-tuning or alignment and can be integrated with open-source fine-tuning libraries. Specifically, we compare against SafeInstruct (Bianchi et al., 2024), a *fine-tuning-stage defense*, as well as RESTA (Bhardwaj et al., 2024) and SafeLora (Hsu et al., 2025), both *post-fine-tuning-stage defenses*. We refer to Appendix D for detailed baseline configurations and intermediate results.

**SafeMERGE.** We merge harmful fine-tuned with *safety-fine-tuned* LoRA layers *only* where the former fails the cosine similarity test based on the threshold $\tau$. Similar to SafeLoRA, we use base and chat/instruct models to define the safety subspace. We explore different weightings, including those from RESTA, as well as balanced combinations summing to 1.0 (e.g., [0.9, 0.1] to [0.5, 0.5]). We additionally explore DARE (Yu et al., 2024) and TIES (Yadav et al., 2023a) merging strategies, but find linear merging sufficient as outlined in Appendix E.4.

## 4 RESULTS AND DISCUSSIONS

We now discuss how **SafeMERGE** competes against the baselines, where we mainly focus on *linear merging* (equation 1). We also discuss some primary ablation results, but defer the majority to Appendix E which includes discussions on tuning weights and selecting the similarity threshold.

**Overall Performance.** Table 1 summarizes the results for all methods. SafeMERGE matches or exceeds utility while significantly reducing harmful outputs. On Llama-2 (GSM8K), it retains near-best accuracy (26.96%) while cutting down DirectHarm and HexPhi rates to 7.50% and 5.70% respectively (from 27.80% and 16.40%). Similarly, on Qwen-2 (GSM8K), SafeMERGE maintains over 72% accuracy with the lowest harmful rates among post–fine-tuning defenses. SafeMERGE achieves this with selective merging—only 34 LoRA layer components (e.g., including Q and V self-attention layer projections) for Qwen-2 (GSM8K) and 28 for Llama-2 (PubMedQA). We provide additional scatter plots and comparisons in Appendix E.1, as well as ablations on SafeMERGE thresholds and weightings in Appendices E.2 and E.3.

Table 1: SafeMERGE performance compared to baselines (SafeInstruct, RESTA, SafeLoRA) for Llama-2-7B-Chat and Qwen-2-7B-Instruct models, finetuned on GSM8K and PubMedQA. Harmfulness (lower is better) is measured by DirectHarm and HexPhi benchmarks.

| Model | Benchmark | Original | Fine-tuned | SafeInstruct | RESTA | SafeLoRA | SafeMERGE |
|---|---|---|---|---|---|---|---|
| Llama-2-7B-Chat (GSM8K) | GSM8K (↑) | 22.67 | 27.37 | 26.00 | 24.94 | 26.15 | **26.96** |
| | DirectHarm (↓) | 5.00 | 27.80 | 7.50 | 7.50 | 10.20 | **7.50** |
| | HexPhi (↓) | 2.00 | 16.40 | 6.20 | **4.30** | 6.90 | 5.70 |
| Llama-2-7B-Chat (PubMedQA) | PubMedQA (↑) | 55.20 | 72.60 | 71.20 | 57.10 | 71.40 | **72.20** |
| | DirectHarm (↓) | 5.00 | 12.50 | 12.20 | **5.80** | 10.70 | 8.10 |
| | HexPhi (↓) | 2.00 | 6.20 | 6.30 | **4.20** | 5.90 | 4.30 |
| Qwen-2-7B-Instruct (GSM8K) | GSM8K (↑) | 58.38 | 70.13 | 72.69 | 60.73 | 74.37 | **72.90** |
| | DirectHarm (↓) | 18.20 | 25.30 | 13.70 | 18.80 | 22.30 | **8.20** |
| | HexPhi (↓) | 11.50 | 16.80 | 9.50 | 15.80 | 14.80 | **7.50** |
| Qwen-2-7B-Instruct (PubMedQA) | PubMedQA (↑) | 73.60 | 79.60 | 80.00 | 75.80 | 82.80 | **80.30** |
| | DirectHarm (↓) | 18.20 | 26.00 | 12.50 | 18.50 | 19.50 | **8.50** |
| | HexPhi (↓) | 11.50 | 13.20 | 5.90 | 14.80 | 14.50 | **5.90** |

**Baseline Comparison.** For fairness, we tune hyperparameters for SafeInstruct, RESTA, and SafeLoRA, selecting the best configurations (see Appendix D). Across Qwen-2-7B-Instruct on GSM8K and PubMedQA, SafeMERGE achieves the highest utility and lowest harmfulness, surpassing vanilla fine-tuning and SafeInstruct while further reducing harmful outputs. On Llama-2-7B-Chat (GSM8K), SafeMERGE attains the best utility and lowest DirectHarm, while RESTA minimizes HexPhi at the cost of utility, which remains close to the base model. On Llama-2-7B-Chat (PubMedQA), SafeMERGE achieves the highest utility, with harmfulness scores close to the instruction-tuned model. While RESTA reduces harmfulness the most, it suffers from significantly lower utility. SafeLoRA preserves utility but struggles to reduce harmfulness, especially on Pub-MedQA, where its Direct-Harm reduction (12.50% to 10.70%) lags behind SafeMERGE (12.50% to 8.10%). A similar trend appears on Qwen-2-7B-Instruct, suggesting simple projection is inefficient for realigning unsafe layers. SafeMERGE addresses this through selective merging, ensuring the best trade-off between safety and performance across all baselines.

**Ablations (Thresholds, Weights, Merging).** In Appendix E.2, we examine the impact of the cosine similarity threshold $\tau$ in SafeMERGE on utility and harmfulness. As $\tau$ increases, more layers are merged, progressively enhancing safety. We also study the role of merging weights in Appendix E.3, finding a *sweet spot* that often maximizes utility while keeping harmfulness low (Figure 2). Additionally, we analyze different merging techniques in Appendix E.4. DARE merging performs similarly to standard linear merging whereas TIES merging is inconsistent, performing well on Llama-2-7B-Chat (GSM8K) but poorly on Llama-2-7B-Chat (PubMedQA) and across datasets for Qwen-2-7B-Instruct, essentially reverting to baseline performance.

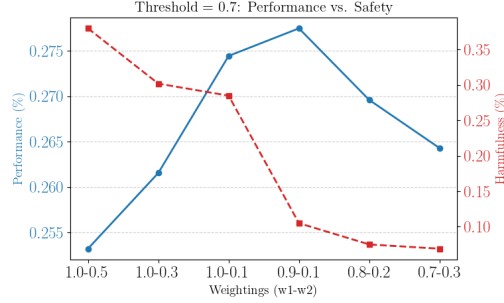

Figure 2: SafeMERGE performance for Llama-2-7B-Chat on GSM8K with threshold 0.5 for different linear weighting combinations. Details in Appendix E.3.

## 5 CONCLUSION

In this paper, we propose **SafeMERGE**, a model-agnostic post-fine-tuning framework for realigning safety. Unlike existing methods for restoring safety after fine-tuning, SafeMERGE identifies and merges only layers with safety degradation, rather than the entire model, by selectively updating LoRA weight adjustments with safety-aligned model layers. Evaluations on Llama-2-7B-Chat and Qwen-2-7B-Instruct across GSM8K, PubMedQA, DirectHarm, and HexPhi demonstrate that SafeMERGE consistently outperforms baselines, achieving best-in-class utility-versus-safety trade-offs. Our results highlight layer-wise selective merging as an effective approach to maintain safety in fine-tuned LLMs without sacrificing performance.

ACKNOWLEDGMENTS

Aladin Djuhera and Holger Boche acknowledge the support of the German Federal Ministry of Education and Research (BMBF) under the program "Souverän. Digital. Vernetzt." as part of the research hubs 6G-life (Grant 16KISK002), QD-CamNetz (Grant 16KISQ077), QuaPhySI (Grant 16KIS1598K), and QUIET (Grant 16KISQ093).

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

## A  RELATED WORK

Recent literature features numerous defenses to preserve or restore *safety alignment* in fine-tuned LLMs. We refer the reader to Yao et al. (2024) for a broad survey, while here we discuss representative methods along three stages of intervention:

**Alignment Stage Defenses.**  These solutions aim to make the base model maximally resilient *before* any user-led fine-tuning. Techniques include large-scale data filtering and alignment procedures (e.g., RLHF (Ouyang et al., 2022)) to prevent harmful adaptation. Representative methods are Vaccine (Huang et al., 2024d), RepNoise (Rosati et al., 2024), CTRL (Liu et al., 2024), TAR (Tamirisa et al., 2024), and Booster (Huang et al., 2024b), which introduce perturbations, adversarial training, or safety constraints to reinforce alignment robustness before fine-tuning.

**Fine-Tuning Stage Defenses.**  These defenses integrate alignment measures *during* fine-tuning. A common approach is to mix safety data into training, as in SafeInstruct (Bianchi et al., 2024) and VLGuard (Zong et al., 2024), or to apply regularization to safety-anchor model outputs, such as LDIFS (Mukhoti et al., 2024), Constrained-SFT (Qi et al., 2024a), and Freeze methods (Wei et al., 2024; Li & Kim, 2025). Additionally, prompt-based safeguards like BEA (Wang et al., 2024) and PTST (Lyu et al., 2025) embed safety triggers into prompts to reinforce alignment without modifying model weights. Some of these methods require explicit adjustments to the fine-tuning pipeline, potentially impractical for black-box fine-tuning.

**Post-Fine-Tuning Stage Defenses.**  Post-training solutions realign a model *after* it has been (potentially unsafely) fine-tuned. This is appealing in scenarios where controlling or monitoring the fine-tuning is unfeasible. Notable examples include SafeLoRA (Hsu et al., 2025), which projects LoRA updates onto a safety subspace derived from a pre-aligned reference model, thereby discarding harmful directions, and RESTA (Bhardwaj et al., 2024), which negatively merges a harmful task vector into a compromised model to restore safe behaviors. Other methods include SOMF (Yi et al., 2024), which utilizes masking techniques to realign a fine-tuned model via task vectors, and Antidote (Huang et al., 2024a), which zeroes out specific harmful weight coordinates to remove undesired responses. These techniques are particularly useful for fine-tuning-as-a-service scenarios, as they can be applied post-hoc with minimal computational cost.

Our method, **SafeMERGE**, fits into this post-training paradigm, specifically drawing from SafeLoRA and RESTA, but takes a more selective, *layer-wise* approach. Instead of globally projecting or adding a single safety vector, SafeMERGE fuses only those LoRA layers whose updates deviate significantly from safety. By preserving benign layers intact, it achieves a stronger trade-off between retaining fine-tuned capabilities and restoring alignment. Numerous other defenses exist, but a comprehensive comparison is beyond the scope.

## B  FINE-TUNING CONFIGURATIONS

### B.1  UTILITY FINE-TUNING

We fine-tune Llama-2-7B-Chat and Qwen-2-7B-Instruct using Llama-Factory (Zheng et al., 2024) with FSDP on $8 \times$ NVIDIA A100 80GB GPUs with the configurations detailed in Table 2.

### B.2  SAFETY FINE-TUNING

We similarly fine-tune the safety model on 100, 500, 1000, and 2500 samples from Bianchi et al. (2024)'s collection using the LoRA parameters from Table 2 with batch size 32, learning rate $1 \times 10^{-4}$, and linear scheduling for 10 epochs each. We select the best (i.e. safest) model (see Table 3).

Table 2: Hyperparameters for GSM8K and PubMedQA fine-tuning across Llama-2 and Qwen-2.

| Parameter | GSM8K | PubMedQA |
|---|---|---|
| Batch Size | 32 | 64 |
| Learning Rate | $1 \times 10^{-4}$ | $1 \times 10^{-4}$ |
| Epochs | 6 | 2 |
| Warmup | 64 steps | 1% of total steps |
| LR Scheduler | Linear | Cosine |
| Weight Decay | 0 | 0.01 |
| LoRA Modules | [q_proj, v_proj] | [q_proj, v_proj] |
| LoRA Rank | 8 | 8 |
| LoRA Alpha | 16 | 16 |
| LoRA Dropout | 0 | 0 |

Table 3: Safety model harmfulness scores (lower is better) for Qwen-2 and Llama-2 for different safety data samples.

| Safety Samples | Llama-2-7B-Chat | | Qwen-2-7B-Instruct | |
|---|---|---|---|---|
| | DirectHarm | HexPhi | DirectHarm | HexPhi |
| 100 samples | 3.00 | 2.30 | 15.50 | 9.90 |
| 500 samples | 1.80 | 2.60 | **6.80** | **3.30** |
| 1000 samples | **1.30** | **1.00** | 7.50 | 3.00 |
| 2500 samples | 1.50 | 2.00 | 9.20 | 6.90 |

## C EVALUATION SETUP

### C.1 UTILITY EVALUATIONS

We assess model performance using EleutherAI's lm-eval-harness framework (Gao et al., 2024) to evaluate utility on GSM8K (0-shot) and PubMedQA benchmarks.

### C.2 SAFETY EVALUATIONS

For safety evaluations, we perform inference on the fine-tuned models and generate responses to harmful prompts from DirectHarm (Lyu et al., 2024) and HexPhi (Qi et al., 2024b). The chosen inference parameters are listed in Table 4. These potentially harmful responses are then evaluated by Llama-Guard-3-8B (Llama Team, 2024) using Meta's moderation pipeline which categorizes outputs into predefined hazard categories (see Table 4).

Table 4: Inference parameters for harmful prompt generation and hazard categories employed by Llama-Guard-3-8B.

| Parameter | Value |
|---|---|
| max_new_tokens | 512 |
| top_p | 1.0 |
| top_k | 0 |
| temperature | 1.0 |
| repetition_penalty | 1.0 |
| length_penalty | 1 |
| batch_size | 1 |

| Category | Description |
|---|---|
| S1 | Violent Crimes |
| S2 | Non-Violent Crimes |
| S3 | Sex-Related Crimes |
| S4 | Child Sexual Exploitation |
| S5 | Defamation |
| S6 | Specialized Advice |
| S7 | Privacy |
| S8 | Intellectual Property |
| S9 | Indiscriminate Weapons |
| S10 | Hate |
| S11 | Suicide & Self-Harm |
| S12 | Sexual Content |
| S13 | Elections |
| S14 | Code Interpreter Abuse |

# D BASELINE CONFIGURATIONS AND RESULTS

## D.1 SAFEINSTRUCT

Following Bianchi et al. (2024), we randomly interleave a set of their harmful Q&A pairs (with safe answers) into the fine-tuning dataset without additional system prompts. We experiment with 100, 500, 1000, and 2500 interleaved safety samples. Since the total number of safety samples remains relatively small (e.g., at most 1.2% of PubMedQA and 28% of GSM8K), we retain the original downstream task fine-tuning hyperparameters.

### D.1.1 FINE-TUNING RESULTS

In general, we confirm Bianchi et al. (2024)'s observation that more samples increase safety, and even may increase utility, at least for our experiments (see Table 5). For comparison with Safe-MERGE, we select the safest variant, i.e. the one with all 2500 safety samples.

Table 5: Comparison of SafeInstruct at various safety sample sizes on Llama-2 and Qwen-2.

| SafeInstruct Number of Samples | Llama-2-7B-Chat | | | | | | Qwen-2-7B-Instruct | | | | | |
| | GSM8K | | | PubMedQA | | | GSM8K | | | PubMedQA | | |
| | HexPhi | DirectHarm | Utility | HexPhi | DirectHarm | Utility | HexPhi | DirectHarm | Utility | HexPhi | DirectHarm | Utility |
| 100 | 10.50 | 10.20 | 23.42 | 10.90 | 26.00 | 69.40 | 10.90 | 19.50 | 71.42 | 6.30 | 15.50 | 79.20 |
| 500 | 7.90 | 10.00 | 23.80 | 10.50 | 18.80 | 69.70 | 10.20 | 17.50 | 72.07 | 5.90 | 14.20 | 79.60 |
| 1000 | 6.80 | 7.90 | 25.17 | 6.90 | 15.20 | 71.20 | 9.90 | 15.70 | 72.42 | 5.90 | 13.50 | 79.20 |
| 2500 | **6.20** | **7.50** | **26.00** | **6.30** | **12.20** | **71.20** | **9.50** | **13.70** | **72.69** | **5.90** | **12.50** | **80.00** |

### D.1.2 UTILITY-VS-PERFORMANCE TRADE-OFF

In Figures 3 and 4, we compare utility (blue, left $y$-axis) and HexPhi harmfulness (red, right $y$-axis) against the number of safety samples for GSM8K experiments, to capture the trade-offs between task utility and harmfulness observed in our study, corroborating Bianchi et al. (2024)'s observations.

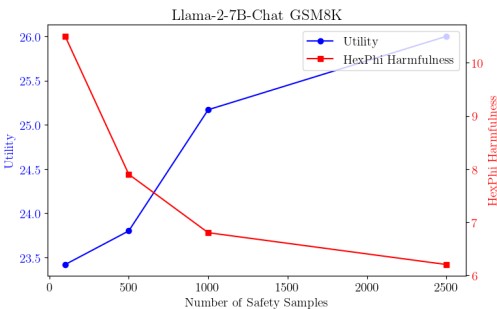

Figure 3: SafeInstruct for Llama-2-7B-Chat (GSM8K, HexPhi)

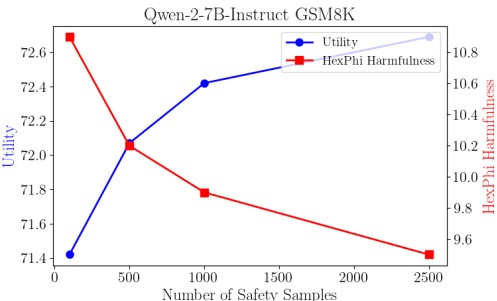

Figure 4: SafeInstruct for Qwen-2-7B-Instruct (GSM8K, HexPhi)

## D.2 RESTA

RESTA (Bhardwaj et al., 2024) constructs a safety vector by fine-tuning a model on harmful data and negating the resulting LoRA parameters. Since the original dataset used in Bhardwaj et al. (2024) is unavailable, we replicate RESTA using AdvBench (Zou et al., 2023) and HarmfulQA (Bhardwaj & Poria, 2023). We evaluate both linear merging and DARE-linear merging, exploring densities from 0.1 to 0.5, and weightings from [1.0, 0.5] to [1.0, 0.1] in addition to ranges that sum up to one (see E.3.

### D.2.1 IMPLEMENTATION

The RESTA methodology follows these steps:

1. Fine-tune a harmful model using AdvBench/HarmfulQA.

2. Negate any part of LoRA weights (e.g. LoRA-B) of the harmful model:

$$W_{\text{harm}}^{\text{LoRA}_B} = -W_{\text{harm}}^{\text{LoRA}_B}$$

3. Merge the negated weights with the original fine-tuned model $\theta_{\text{SFT}}^o$:

$$\theta_{\text{merged}} = \theta_{\text{SFT}}^o + \alpha \cdot \theta_{\text{harmful}}$$

   where $\alpha$ is the weighting factor.

4. Apply DARE rescaling if desired.

We implement LoRA merging using HuggingFace's PEFT library, which supports linear and DARE-linear adapter merging.

### D.2.2 HARMFUL FINE-TUNING

We fine-tune both Llama-2-7B-Chat and Qwen-2-7B-Instruct models on harmful AdvBench and HarmfulQA datasets with a batch size of 32, learning rate of $1 \times 10^{-4}$, and linear scheduling for 5 epochs, respectively. We report harmfulness scores across DirectHarm and HexPhi in Table 6:

Table 6: Harmful model scores (higher is better) for Llama-2-7B-Chat and Qwen-2-7B-Instruct for AdvBench and HarmfulQA.

| | AdvBench | | HarmfulQA | |
|---|---|---|---|---|
| **Model** | DirectHarm | HexPhi | DirectHarm | HexPhi |
| Llama-2-7B-Chat | 38.30 | 36.50 | 94.00 | 97.40 |
| Qwen-2-7B-Instruct | 59.50 | 47.70 | 72.00 | 76.00 |

### D.2.3 RESTA WEIGHTING VS. DARE-LINEAR DENSITY

We analyze the trade-off between utility and harmfulness using linear merging and DARE-linear merging. The results for Qwen-2-7B-Instruct fine-tuned on AdvBench and HarmfulQA are shown in below Figures 5a, 5b, 5c, and 5d. We see that DARE-linear merging consistently leads to better safety scores for PubMedQA but at the cost of lower task performance. Thus, RESTA alone is insufficient to restore safety while retaining utility, at least for the utilized AdvBench and HarmfulQA datasets in our experiments.

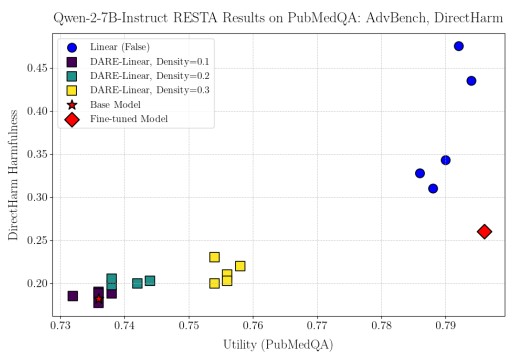

(a) AdvBench: Utility vs. DirectHarm Harmfulness.

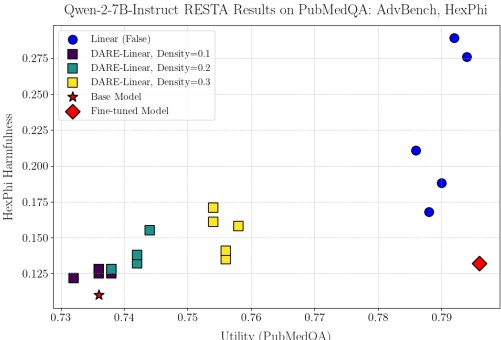
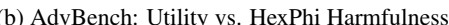

(b) AdvBench: Utility vs. HexPhi Harmfulness.

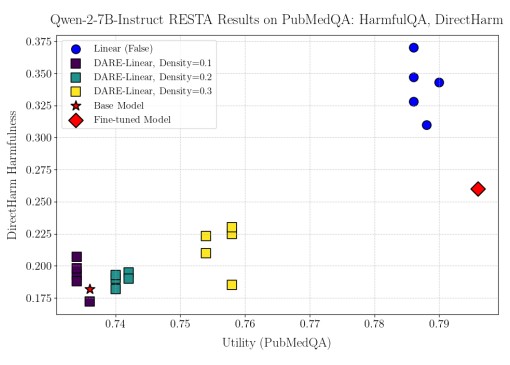

(c) HarmfulQA: Utility vs. DirectHarm Harmfulness.

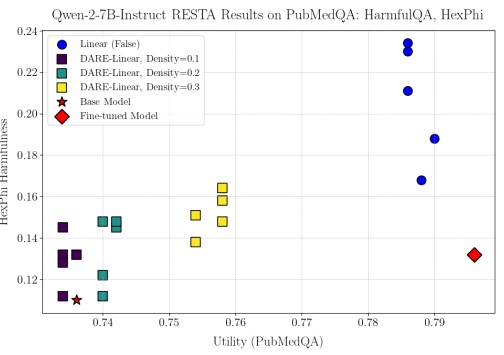

(d) HarmfulQA: Utility vs. HexPhi Harmfulness.

Figure 5: Comparison of RESTA's utility vs. harmfulness trade-off for Qwen-2-7B-Instruct models across different datasets. The first row shows results for AdvBench, while the second row presents results for HarmfulQA. Each row contains DirectHarm on the left and HexPhi on the right.

## D.3   SAFELORA

SafeLoRA (Hsu et al., 2025) mitigates safety degradation in fine-tuned models by projecting LoRA weight updates onto a safety-aligned subspace. The projection matrix is constructed using an unaligned base model and a safety-aligned instruct model. We apply SafeLoRA to Llama-2 and Qwen-2 using base and instruct variants, tuning cosine similarity thresholds $\tau$ between 0.1 and 1.0.

### D.3.1   IMPLEMENTATION

For SafeLoRA, we follow the methodology and repository from Hsu et al. (2025). The projection matrix is computed using Llama-2-7B-Chat and Llama-2-7B (base). We find that Llama-2-7B-Chat is already well safety-aligned, making additional safety finetuning unnecessary. We also investigate two projection approaches for Qwen-2-7B: (i) Base Model Projection: Using Qwen-2-7B (base), (ii) Safety Model Projection: Using a safety-tuned model (500 samples). Results however show that most LoRA projections remain identical across both projection methods.

### D.3.2   THRESHOLD SELECTION AND MERGED LAYERS

We analyze the threshold factor and the number of projected layers in SafeLoRA. In general lower thresholds result in less projected layers, preserving performance but limiting safety improvements. Due to the LoRA formulation, projection is only required for either LoRA-A or LoRA-B, as multiplication with the other counterpart includes the projection inherently. Thus, the maximum number of projected layers is 56 for Qwen-2 models and 64 for Llama-2 models. Below Figure 6 shows the progression plot for Llama-2-7B-Chat, finetuned on GSM8K.

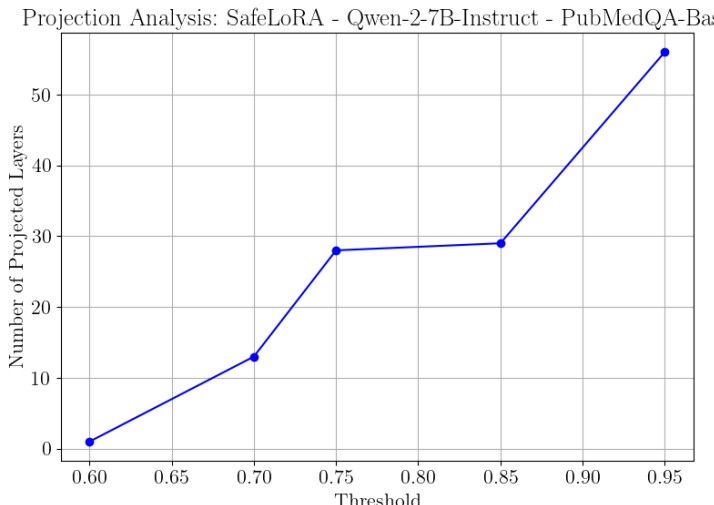

Figure 6: SafeLoRA: threshold progression vs. number of projected LoRA layers for Llama-2-7B-Chat (GSM8K).

### D.3.3 PROJECTION VS. HARMFULNESS VS. UTILITY

We compare SafeLoRA's performance against the number of projected layers and harmfulness benchmarks for Llama-2-7B-Chat (GSM8K) in below Figure 7. In general, SafeLoRA preserves task performance similar to SafeInstruct, however, reduces harmfulness less effectively than Safe-Instruct or SafeMERGE. Since not all LoRA layers are projected, SafeLoRA retains higher utility on challenging datasets like GSM8K for Llama-2 and PubMedQA for Qwen-2. SafeMERGE is motivated by these findings, where layer-wise merging balances safety and performance more effectively.

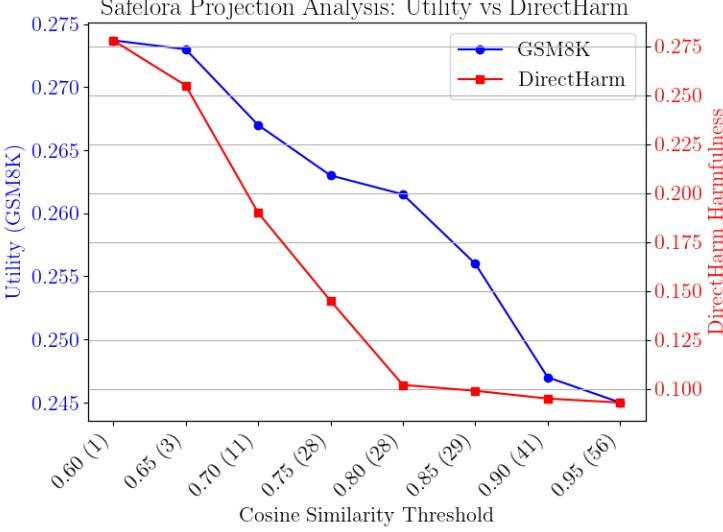

Figure 7: SafeLoRA: projection vs. harmfulness vs. utility for Llama-2-7B-Chat (GSM8K).

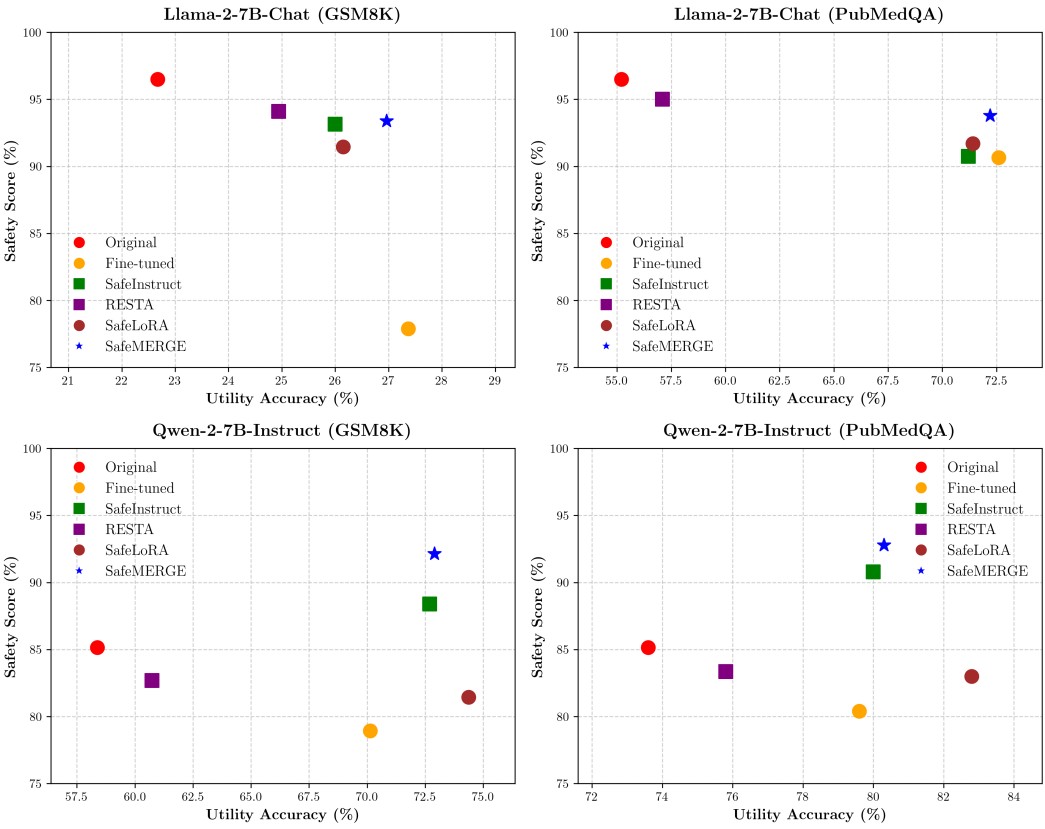

Figure 8: SafeMERGE performance against baselines. Safety is reported as the mean score between DirectHarm and HexPhi benchmarks.

# E    SAFEMERGE RESULTS

## E.1    SAFEMERGE PERFORMANCE AGAINST BASELINES

We present a detailed scatter plot in Figure 8, positioning SafeMERGE against other baselines. Safety is computed as a mean score using the formula:

$$\text{Safety Score} = \frac{(100 - d) + (100 - h)}{2}$$

where $d$ represents DirectHarm harmfulness and $h$ represents HexPhi harmfulness.

We observe that SafeMERGE consistently outperforms the baselines, achieving a superior balance between safety and performance.

## E.2    THRESHOLD PROGRESSION: MERGED LORA LAYERS VS. FINETUNING PERFORMANCE VS. HARMFULNESS

We show the progression of merged LoRA layers vs. fine-tuning performance vs. harmfulness (DirectHarm) in Figures 9 and 10 for different thresholds in Llama-2 and Qwen-2 models on GSM8K, respectively. We observe that merging all LoRA layers, i.e. $\tau = 1$, converges to the performance of full linear model merging. We also observe that merging as few as 8 layers already leads to a significant decrease in harmfulness.

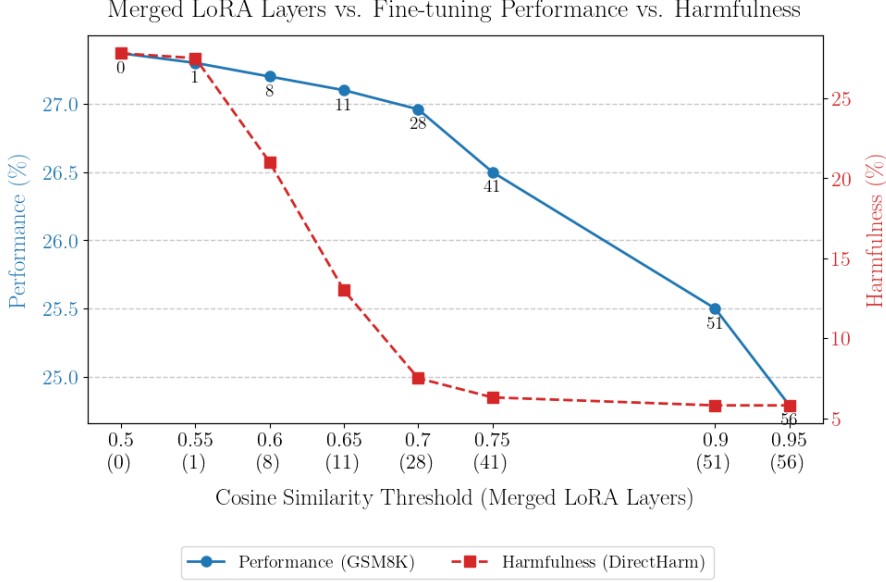

Figure 9: Llama-2-7B-Chat (GSM8K, DirectHarm). SafeMERGE performance with weighting $[0.8, 0.2]$ for different cosine similarity thresholds. A threshold of 0 (leftmost point) indicates no merging, i.e. the baseline. Increasing the threshold increases the number of merged layers and thus converges to full linear model merging performance in both task utility and harmfulness.

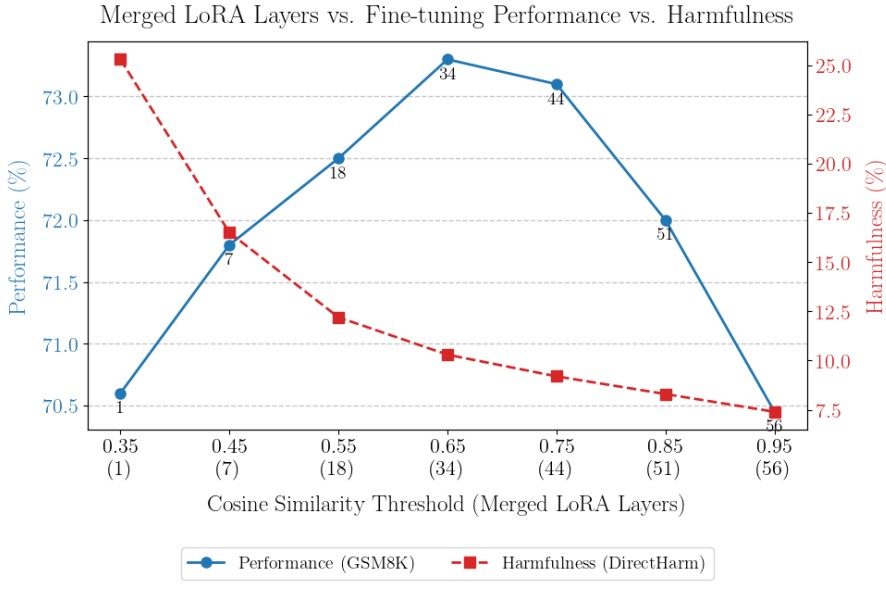

Figure 10: Qwen-2-7B-Instruct (GSM8K, DirectHarm). SafeMERGE performance with weighting $[0.7, 0.3]$ for different cosine similarity thresholds. A threshold of 0 (leftmost point) indicates no merging, i.e. the baseline. Increasing the threshold increases the number of merged layers and thus converges to full linear model merging performance in both task utility and harmfulness.

### E.3 IMPACT OF DIFFERENT WEIGHTINGS IN SAFEMERGE FOR A GIVEN THRESHOLD

We show the impact of different linear weighting combinations for a given threshold in Figure 11 for Llama-2 and Qwen-2 models (GSM8K). We observe optimal trade-offs between safety and downstream task performance for weightings that sum up to 1.0, often observing a sweet spot around ranges between $[0.9, 0.1]$ to $[0.6, 0.4]$.

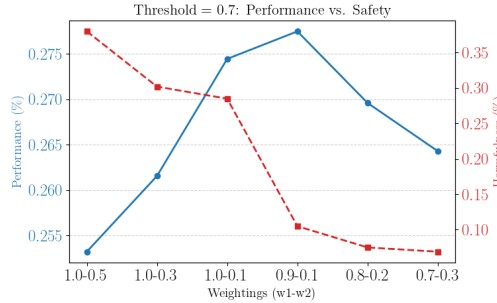
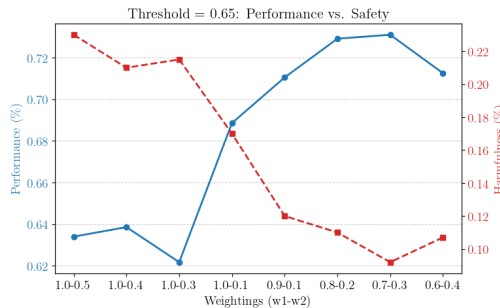

(a) SafeMERGE performance for Llama-2-7B-Chat on GSM8K with threshold 0.5 for different weighting combinations. Best results are achieved when weights sum up to 1.0 during linear merging. In general, increasing the safe model's contribution increases safety at the price of downstream task performance.

(b) SafeMERGE performance for Qwen-2-7B-Instruct on GSM8K with threshold 0.65 for different weighting combinations. Best results are achieved when weights sum up to 1.0 during linear merging. In general, increasing the safe model's contribution increases safety at the price of downstream task performance.

Figure 11: SafeMERGE: impact of different weightings in SafeMERGE for a given threshold.

### E.4 IMPACT OF DIFFERENT MERGING STRATEGIES

We report utility and safety benchmarks in Table 7 for both Llama-2 and Qwen-2 on GSM8K and PubMedQA, comparing linear, DARE-linear, and TIES merging strategies. We observe that linear and DARE-linear merging yield similar results, with no significant deviations between them. However, TIES merging leads to inconsistencies. On Llama-2 (GSM8K), it improves safety compared to linear and DARE-linear merging while maintaining competitive utility. Yet, in all other experiments, TIES merging degrades model performance, reverting it toward baseline levels and, in some cases, even increasing harmfulness. This suggests that TIES merging fails to suppress harmful directions and may inadvertently reinforce them during layer-wise merging. A deeper analysis of this behavior is warranted and left for future work.

Table 7: SafeMERGE performance for Linear, DARE-Linear, and TIES merging strategies.

| | Llama-2-7B-Chat | | | | | | Qwen-2-7B-Instruct | | | | | |
|---|---|---|---|---|---|---|---|---|---|---|---|---|
| | GSM8K | | | PubMedQA | | | GSM8K | | | PubMedQA | | |
| **Merging Strategy** | HexPhi | DirectHarm | Utility | HexPhi | DirectHarm | Utility | HexPhi | DirectHarm | Utility | HexPhi | DirectHarm | Utility |
| Linear | 5.70 | 7.50 | 26.96 | 4.30 | 8.10 | 72.20 | 7.50 | 8.20 | 72.90 | 5.90 | 8.50 | 80.30 |
| DARE-Linear | 5.70 | 8.10 | 26.80 | 4.50 | 7.90 | 72.4 | 7.50 | 8.30 | 72.60 | 5.30 | 8.30 | 79.90 |
| TIES | 4.60 | 5.80 | 26.46 | 3.30 | 4.30 | 55.20 | 12.50 | 15.80 | 60.73 | 13.80 | 18.50 | 75.40 |
| Original | 2.00 | 5.00 | 22.67 | 2.00 | 5.00 | 55.20 | 11.50 | 18.20 | 58.38 | 11.50 | 18.20 | 73.60 |

