# OpenReview forum: "SafeMERGE: Preserving Safety Alignment in Fine-Tuned Large Language Models via Selective Layer-Wise Model Merging"
_ICLR.cc/2025/Workshop/BuildingTrust — BuildingTrust_

### Official Review · Reviewer_xW8j · 2025-02-18
**The authors propose a novel mechanism and provide justification.**

**Rating:** 7
**Confidence:** 4

**Review:**

- The authors propose a mechanism to update the model layer-wise in a manner that the output of the cosine similarity of the LLM would not deviate from the original one.
- The rationale behind the method is clear.
- The paper is well-written and easy to follow.

---

### Official Review · Reviewer_nn6B · 2025-02-28
**Building on SafeLoRA, the paper claims improved LLM safety alignment, could benefit from expanded benchmarking and better formatting.**

**Rating:** 6
**Confidence:** 3

**Review:**

# Significance

- This paper builds upon SafeLoRA and introduces SafeMERGE, to maintain safety alignment in fine-tuned LLMs.
- The key distinction between SafeMERGE and SafeLoRA is that _SafeMERGE merges unsafe layers with safe layers after identifying them via the safety-aligned subspace, whereas SafeLoRA projects unsafe layers onto the safety-aligned subspace_.
- Fine-tuning LLMs on downstream tasks can unintentionally degrade their safety alignment, even when using benign datasets. This issue is highly relevant to the workshop community.
- The authors claim that SafeMERGE achieves a better trade-off between utility and safety compared to existing baselines.

# Overall Quality and Evaluation

- The paper presents comprehensive experimental results using widely used open-weight LLMs (LLaMA-2-7B-Chat and Qwen-2-7B-Instruct) and benchmark datasets (GSM8K and PubMedQA).
- The evaluation methodology is appropriate, using:
  - Exact-match accuracy for GSM8K and classification accuracy for PubMedQA to measure utility.
  - DirectHarm and HexPhi datasets with Llama-Guard-3-8B to assess model safety.
- SafeMERGE is compared against relevant baselines (SafeInstruct, RESTA, SafeLoRA) with well-tuned hyperparameters to ensure fairness.
- The paper includes ablation studies to analyse key components of SafeMERGE, such as merging strategies, weighting schemes, and similarity thresholds. The appendices provide additional technical details.

# Suggestions for Improving the Paper

-  The decision to submit a short 4-page paper while placing critical content (e.g., related work, benchmarking experiments) in the appendix limits the clarity and coherence of the work. Given that the workshop allows a 9-page format, expanding the paper could help present the research in a more structured and complete manner.
- Comparing LLaMA-2-7B-Chat on the same benchmark datasets used in the SafeLoRA paper would improve reproducibility and allow for a clearer evaluation of SafeMERGE’s improvements.
- While the paper references the use of harmful prompt–safe response pairs, a more detailed description of this data (e.g., the nature of harmful prompts and the methodology for creating safe responses) would enhance reproducibility and transparency.
- Safety threshold τ: Since the paper highlights the importance of the safety threshold τ, a more in-depth discussion on how to determine an appropriate τ for different models and tasks would be beneficial.

---

### Official Review · Reviewer_NYLk · 2025-03-01
**Review of "SafeMERGE: Preserving Safety Alignment in Fine-Tuned Large Language Models via Selective Layer-Wise Model Merging"**

**Rating:** 6
**Confidence:** 3

**Review:**

The paper proposes a method to retain safety alignment after fine-tuning an LLM on task-specific data. It achieves this by selectively replacing the fine-tuned LoRA weights of certain layers—specifically, those that deviate from the safety-aligned subspace—with linear combinations of the fine-tuned and aligned weights. This results in models that are both less harmful and still highly performant.

**Positive Aspects:**
- Effective and practical method for maintaining safety alignment after fine-tuning.
- Thorough experimental evaluation.
- Comparison to several baselines, with the proposed method mostly outperforming them.

**Negative Aspects:**
- Some explanations could be more detailed. In particular, the presentation of the safety-aligned subspace does not seem well-motivated. While I understand that it originates from a previous paper, some additional explanation would be helpful for many readers.
- If I understand correctly, the proposed method introduces additional hyperparameters—the merging factors—which is tuned. This raises concerns about fairness when comparing to baselines with fewer degrees of freedom. Perhaps the merging factors should be fixed across all models and datasets to ensure a more equitable comparison.

**Additional Comments:**
- In many figures (e.g., Figures 2, 9, 10, and 11), including the scores of the purely aligned and purely fine-tuned models for comparison would make it easier to interpret the results.
- DARE merging is mentioned multiple times, but no source is cited for it. Not all readers will be familiar with it, so a citation should be included.
- Regarding linear merging, the paper frequently emphasizes that the factors for the fine-tuned and aligned models summing to one is empirically the most successful approach. However, this seems like a rather obvious choice, so I’m not sure if it warrants such strong emphasis.

---

### Decision · Program_Chairs · 2025-03-01

Accept